# Neonatal Hyperglycemia and Neurodevelopmental Outcomes in Preterm Infants: A Review

**DOI:** 10.3390/children9101541

**Published:** 2022-10-09

**Authors:** Silvia Guiducci, Leonardo Meggiolaro, Anna Righetto, Marco Piccoli, Eugenio Baraldi, Alfonso Galderisi

**Affiliations:** 1Departement of Woman’s and Child’s Health, University of Padova, 35128 Padova, Italy or; 2Hôpital Necker-Enfants Malades, 75015 Paris, France

**Keywords:** neonatal glucose, hyperglycemia, preterm infant, neurodevelopmental outcome, neurodevelopment impairment

## Abstract

Glucose impairment is common in preterm infants but the impact of early neonatal hyperglycemia on long term neurodevelopment is still highly controversial. This review reports current evidence of the effect of hyperglycemia on neurodevelopmental outcome. It was conducted according to the PRISMA guidelines. We searched MEDLINE via PubMed; EMBASE via Ovid; the Cochrane Central Register of Controlled Trials; the Cochrane Library; ClinicalTrials.gov; and the World Health Organization’s International Trials Registry and Platform. We included studies that investigated the association between hyperglycemia, defined as at least one episode of glycemia ≥8 mmol/L, and neurodevelopment outcome evaluated either through the Griffiths Mental Developmental Scales (GMDS) or the Bayley Scales of Infant Development (BSID) for the first 5 years of life, and the Wechsler Intelligence Scale for Children (WISC) and the Movement Assessment Battery for Children (MABC) for the following age category. We selected six studies, comprising 2226 infants in total and which included 1059 (48%) infants for whom neurodevelopment assessment was available. We found an association between hyperglycemia and neurological delay in the first two years of life, especially for motor functions; this result was confirmed in later childhood. The quality of evidence was poor; therefore, the negative influence of neonatal hyperglycemia on the neurological development of preterm infants must be investigated in further studies.

## 1. Introduction

The incidence of the preterm births is estimated to be 11.1% of all live births worldwide, ranging from a minimum of 5% in European countries to a maximum of 18% in some African countries [1,2,3]. Despite the improvement of perinatal and neonatal care, the incidence of neurodevelopmental impairment in preterm infants is still significant, varying from 38% in preterm <28 weeks [4] to 4–10% in those born at 31–34 weeks [5]. Hyperglycemia is considered a risk determinant of the outcome in this population [6,7]. Indeed, hyperglycemia was demonstrated to be very common among very low birth weight (VLBW) infants [8], with the risk of developing hyperglycemia inversely related to gestational age (GA) and birth weight [9].

The pathogenesis of hyperglycemia in preterm infants is still uncertain but multiple factors are known as possible contributors. First, preterm infants have a poorer insulin secretory response to glucose [10] due to both the reduction of β-cell mass, especially in the case of intrauterine growth restriction, and the immaturity of the β-cells, related to the low expression of glucose transporter 2 (GLUT2) [11]. Moreover, enteral feeding is often delayed in preterm infants, meaning a low production of incretins—the hormones that stimulate insulin secretion [12]. Frequently preterm infants need intravenous glucose infusion but, unlike adults, the endogenous glucose production is not suppressed: this may be derived from both an immature expression of GLUT2 in the liver that causes continuous hepatic glucose production, and a reduced glucose uptake by insulin-sensitive tissue, such as adipose and skeletal muscle, which have a lower GLUT4 [13] expression. This relative insulin resistance may be worsened by increased proinflammatory cytokines, for example due to sepsis, increased counter-regulatory hormones associated to stress—such as epinephrine and cortisol—and intensive care intervention, with the use of inotropes and corticosteroids [14]. Therefore, knowing the possible causes of hyperglycemia would help adopt the best intervention strategy.

Regarding the role of hyperglycemia in outcome, it was associated with increased mortality and morbidity [15], including retinopathy of prematurity [16,17], sepsis [18,19], severe intraventricular hemorrhage and prolonged hospital stay [6]. However, little is documented on the long-term effects of hyperglycemia [20]. We, therefore, explored the association between neonatal hyperglycemia in preterm infants and neurodevelopmental outcomes in early and late childhood.

## 2. Materials and Methods

This review was conducted in accordance with the Preferred Reporting Items for Systematic Reviews and Meta-analysis (PRISMA) statement [21].

### 2.1. Inclusion Criteria

We included randomized and quasi-randomized trials, cohort studies and case control studies. We excluded case reports, case series, conference abstracts and unpublished studies. The population was preterm infant with GA < 37 weeks with glucose monitoring during the first 28 days of life (DOL) and with a neurodevelopment outcome evaluation based on validated tools with no age limits. They were classified as:Hyperglycemic group: infants with at least 1 episode of hyperglycemia defined as blood glucose concentration ≥8 mmol/L treated or not with insulin;Control group: infants without any episodes of hyperglycemia.

We excluded studies including at-term infants, newborns with congenital malformations or inherited metabolic disorders or congenital hyperinsulinism.

The outcome of interest was the presence of neurodevelopmental impairment evaluated through validated tools. The validation tools were the GMDS or the BSID for the first 5 years of life, and the WISC and the MABC for the subsequent age category. We defined neurodevelopmental impairment (NDI) as the presence of any of the follow conditions: a BSID Mental Development Index < −1 Standard Deviation (SD); a GMDS General Quotient < −1 SD; a WISC full Intelligence Quotient < −1 SD; a MABC total score ≤ 5th centile; cerebral palsy; bilateral blindness (visual acuity < 20/60 in better eye or requiring spectacles); and bilateral deafness (requiring aids). We also recorded the presence of executive dysfunctions and emotional–behavioral difficulties, as defined by authors.

We considered a standardized score < −1 SD from the mean of the tools to define NDI, identifying all the possible levels of neurological delay. All the validated tools considered a moderate/severe NDI score to be < −2 SD but more uncertainty existed in the interpretation of scores between −2 SD and −1 SD: this range defined the mild category, but several studies showed a tendency to underestimate NDI, especially the BSID [22,23].

### 2.2. Research Strategy

The electronic databases searched were: MEDLINE via PubMed (1976 to 10 August 2021); EMBASE via Ovid (1974 to 10 August 2021), the Cochrane Central Register of Controlled Trials and the Cochrane Library. We also searched clinical trials databases—ClinicalTrials.gov and the World Health Organization’s International Trials Registry and Platform—for ongoing or recently completed trials on 10 August 2021. 

The search strings included the following terms: Infant, Newborn [Mesh] OR preterm OR infan* OR newborn OR neonat* OR premature OR low birth weight OR very low birth weight OR LBW OR VLBW AND Hyperglycemia [Mesh] OR Blood Glucose [Mesh] OR hyperglycaem* OR hyperglycem* AND neurodevelop* OR outcome OR neurodevelopment outcome OR neurodevelopment*. The search was restricted to studies involving humans and published in English. There was no limit on the year of publication. The search was last updated on August 10, 2021. We also hand-searched bibliographies of included studies, review papers and conference abstracts to identify additional items.

### 2.3. Data Extraction

Two groups of authors conducted the search independently. The records identified for full-text screening were also reviewed by two different groups of authors independently. Conflicts were resolved by consensus or after consultation with a third author. Screening and eligibility assessments were performed using COVIDENCE (http://www.covidence.org/ (accessed on 1 September 2021)).

## 3. Results

### 3.1. Research Strategy

A PRISMA flow chart of screening and selection results is shown in Figure 1. Of 623 records identified through databases and hand searching, 221 were duplicates and were removed. Of the remaining 402 studies, 393 were excluded following title and abstract screening. One further study was excluded following full-text review due to the lack of a full text [24] and two others were excluded due to the lack of control groups [20,25]. Thus, a total of 6 studies were included, comprising 2226 infants at enrolment, with neurodevelopment assessment available in 1059 (48%) infants [19,26,27,28,29,30]. Meta-analysis was not performed due to the heterogeneity of the studies, especially the tools used to assess long-term neurodevelopment outcome. 

### 3.2. Characteristics of Selected Studies

Details of included studies are summarized in Table 1. 

All the included studies were cohort studies: five retrospective and one prospective. All the studies were conducted in developed countries, including Europe, the USA and Oceania. Four studies were conducted in the 2000s and two in the 2010s. 

All the studies included preterm infants ≤32 GA. 

The incidence of hyperglycemia ranged from 8% [19] to 87% [29] during the first four weeks of life. The heterogeneity of the included criteria and protocols to detect glycemia could explain the wide range of incidence.

Van der Lugt et al. [19] conducted a retrospective cohort study; they selected 859 preterm infants ≤32 GA admitted for intensive care and divided them into two groups: exposed to hyperglycemia during hospitalization and unexposed. They defined hyperglycemia as at least 2 blood glucose levels of ≥10 mmol/L during a 12-h period. Their therapeutic strategy consisted of a reduction in glucose intake down to 5–6 mg/kg/min and starting insulin at 0.05 IU/kg/h, if hyperglycemia persisted after these 12 h. The neurological outcome was evaluated at 2 years CA through the Hempel neurological examination and the Child Behavior Checklist/2–3.

Heald et al. [26] conducted a retrospective study: they identified preterm infants <29 GA exposed to hyperglycemia, defined as ≥10 mmol/L and significant glycosuria on 10% dextrose parenteral nutrition. They divided the infants into two groups by insulin therapy—start dose 0.01 IU/kg/h—and conducted a neurological assessment at 1 year CA using the GMDS and BSID-II. 

Tottman et al. [27] conducted a retrospective cohort study: they identified 443 preterm infants with <30 GA or with BW <1500 g with an available glucose profile for the first week of life. Patients were categorized into four groups according to glucose excursion: normoglycemic, hypoglycemic, hyperglycemic and unstable. Hyperglycemia was defined as a blood glucose concentration ≥8.6 mmol/L on ≥2 measures >1 h apart or any >10 mmol/L; treatment consisted of a reduction in the glucose infusion rate or starting insulin therapy to maintain glycemia at 4–10 mmol/L. Neurodevelopmental assessment was conducted at 2 years CA by BSID-II and III.

Villamizar et al. [28] conducted a retrospective post-hoc analysis of data collected prospectively: they included 97 preterm infants with a birth weight <1500 g and appropriate for their gestational age. Hyperglycemia was defined as 1 or more blood glucose measurements >8.3 mmol/L during the first 7 DOL. Therapy consisted of the adjustment of the glucose infusion rate with no mention of insulin. Patients were divided into three groups according to the presence and duration of hyperglycemia: 0 days, 1–4 days, ≥5 days. Neurological outcome was evaluated at 1 year CA by BSID–III comparing the normoglycemic group to the other two categories of hyperglycemic patients.

Zamir et al. [29] conducted a retrospective study: they included 533 preterm infants < 27 GA with all glucose measurements and insulin therapy data available during the first 28 DOL. No standard insulin treatment protocol was adopted. They defined four different categories of hyperglycemia: >8, >10, >12 or >14 mmol/L, occurring at least once or on 2 or 3 consecutive days. Neurodevelopmental assessment was performed at 6.5 years of age by WISC-IV and MABC-II.

Boscarino et al. [30] conducted a prospective study—the only one included in this review: they included 280 preterm infants with <32 GA or with a BW < 1500 g and divided them into two groups: exposed to hyperglycemia, defined as 2 consecutive >10 mmol/L at least 3 h apart, or not exposed to hyperglycemia. They evaluated survival without NDI at 2 years CA with BSID-III

The definition of hyperglycemia was not homogeneous among studies: the maximum range in Zamir et al. was from 8 to 14 mmol/L [29]; three studies used a threshold value of 10 mmol/L [19,26,30]; one study considered two possible thresholds of 8.6 or 10 mmol/L [27]; and Zamir et al. [29] considered four different categories of hyperglycemia—8–10–12–14 mmol/L. Furthermore, there was no agreement on the duration of hyperglycemia: some authors defined hyperglycemia as the finding of a single value over the threshold [26,27], while others considered two or more consecutive values over the threshold [19,27,30]. 

The assay method used to measure blood glucose was different among studies: whole blood glucose in two studies [19,27], plasma glucose in one study [29], capillary glucose in one study [30] and not reported in two studies [26,28]. No study used a subcutaneous sensor for continuous glucose monitoring.

The treatment protocol for hyperglycemia differed between studies even though the initial reduction in parenteral glucose intakes and, subsequently, the beginning of insulin therapy was common: the minimum glucose infusion was reported in three studies as ranging between 5 and 7 mg/kg/min [19,28,30], while the starting dose of insulin was described as being in the range of 0.01–0.5 IU/kg/h in three other studies [19,26,30]. 

### 3.3. Results of Individual Studies

Neurodevelopment outcome is summarized in Table 2. It was assessed at two different ages: in five studies between 1- and 2-years CA [19,26,27,28,30] and in one study at 6.5 years CA [29]. Neurological outcome was available for 1059 (48%) with 541 (51%) infants with a history of neonatal hyperglycemia. 

Validated tools that we considered as outcome measures were used at 1 year CA in two studies using the BSID and/or GMDS [26,28]; at 2 years CA in two studies using the GMDS, BSID II and BSID III [27,30]; at 6.5 years in one study using WISC-IV and MABC-II [29]. Der Lugt et al. used neurological examination by Hempel and the Child Behavior Checklist/2-3 for behavioral assessment at 2 years CA [19]. We did not find studies investigating longer neurological outcomes.

Neurodevelopmental assessment tools used in the studies were different and authors used different definitions of NDI derived from the integration of several clinical aspects, among which the BSID, GMDS, WISC and MABC were considered but were not the sole measure. Furthermore, some studies [27,29] categorized NDI into different levels of seriousness based on the results of these validated tools, where the mild category coincided with a score of < −1 SD and the moderate/severe category with a score of <−2 SD.

We divided the studies into two groups based on the age of follow-up: early childhood—1–5 years CA—where we considered five studies [19,26,27,28,30], and later childhood—6–11 years—that included one study [29].

#### 3.3.1. Early Childhood

Heald et al. [26] compared the neurological outcomes at 1 year CA in preterm infants exposed to insulin therapy due to hyperglycemia (9 patients) or not exposed (64 patients); they used the GMDS for neurological assessment and shifted to BSID–II during the study. They also considered cerebral palsy, defined as the inability to walk without aids, bilateral blindness—visual acuity <6/60—and bilateral deafness—requiring bilateral hearing aids or cochlear implants. The authors did not find a significant difference between the two groups. 

Villamizar et al. [28] evaluated NDI at 1 year CA in a cohort of preterm infants exposed to hyperglycemia or not exposed and dividing hyperglycemic infants into two groups based on the duration of hyperglycemia—1–4 days or ≥5 days. They found that hyperglycemia ≥5 days had a significant negative impact on the BSID–III score for all three domains—*p*-values of 0.02 for cognition and language and a *p*-value of 0.01 for motor function. These results persisted after adjustment for illness severity during the first 2 DOL—evaluated using the mean of the Score for Neonatal Acute Physiology–II—but not for first-week nutrition intake of calories and protein. The latter finding suggested that nutritional intake contributed to the relationship between neonatal hyperglycemia and neurodevelopment.

Concerning the two-year CA follow-up, Van der Lugt et al. [19] compared preterm infants exposed to hyperglycemia and not exposed; they found a significant higher incidence of NDI in the exposed group—a *p*-value of 0.036 for the neurological outcome and a *p*-value of 0.021 for the behavioral outcome—but they did not use either the BSID or the GMDS.

Tottman et al. [27] analyzed neurological outcomes at 2 years CA in preterm infants divided into four groups: normoglycemic, hyperglycemic, hypoglycemic and unstable. Considering only the two groups of hyperglycemic and normoglycemic infants, they found more common NDI in the hyperglycemic group with a low odds of survival without impairment—*p*-value 0.006. However, after adjustment for confounders—GA, birth weight z-score and socioeconomic quintile—there were no significant associations between the different glycemic categories. 

Boscarino et al. [30] compared neurodevelopment assessed using BSID–III at 2 years CA in preterm infants exposed to hyperglycemia, finding that the rate of NDI was higher in the hyperglycemic group in all three domains, with statistically significant differences for the cognition and motor domains—*p*-values < 0.05 and <0.001, respectively. After adjustment for background characteristics, hyperglycemia was confirmed as a risk factor only for motor delay—*p*-value 0.007—together with male sex.

#### 3.3.2. Later Childhood

Zamir et al. [29] compared neurological outcomes at 6.5 years in preterm infants exposed or not to hyperglycemia; they found no significant association with moderate to severe NDI—here defined as a WISC Intelligence Quotient < −2 SD, cerebral palsy, visual acuity <20/63 in the better eye and deafness requiring an aid. However, focusing on the WISC score and MABC, they found a negative association between hyperglycemia >8 mmol/L for ≥2 consecutive days and the WISC Intelligence Quotient—*p*-value 0.045—and between hyperglycemia >8 mmol/L for ≥3 days or any episodes of hyperglycemia >10 mmol/L and a MABC–II—*p*-value < 0.05. These were suggestive of both entity and duration of neonatal hyperglycemia being associated with poorer motor outcomes at 6.5 years and lower intelligence scores.

#### 3.3.3. Quality of Evidence

The quality of evidence was poor due to several limits. First, all the studies were cohort with just one being purely prospective [30] and two involving retrospective analysis of prospectively collected data [28,29]. All the studies had a relatively small sample size and, as previously emphasized, thresholds and durations for defining hyperglycemia were heterogeneous. Moreover, treatment strategies among the studies were different, making a comparison difficult. As regard neurodevelopment outcome, all the studies except Van der Lugt [19], used validated tools but the way data was reported was heterogeneous. The entity of drop out at follow up was quite high but small in two studies—Van der Lugt [19] 11% and Boscarino [30] 38%—and this could make the results less representative. All the studies except Van der Lugt adjusted for confounders; nevertheless, they did not assess the same confounders. 

## 4. Discussion

Preterm infants have a high risk of hyperglycemia [13]. In early childhood—1–5 years CA—neonatal hyperglycemia is associated with neurodevelopmental impairment in preterm infants, especially with respect to the motor domain. This result is confirmed in later childhood—6–11 years CA—by a single observational study [29]. The relationship between neonatal hyperglycemia and neurodevelopmental outcome remains controversial.

First, the definition of hyperglycemia is highly debated: there is no agreement among the examined studies about absolute thresholds and the duration of exposure. The prevailing definition for hyperglycemia was a blood glucose value ≥10 mmol/L for at least two consecutive periods, but the lower thresholds of 8 and 8.2 mmol/L were also investigated. The European Society for Pediatric Gastroenterology, Hepatology and Nutrition (ESPGHAN) recommends a lower threshold of 8 mmol/L [31], and this threshold was confirmed by a recent review [15] of the role of hyperglycemia in mortality and early morbidities. 

Analysis of the neurodevelopmental outcomes of the included studies shows that both the higher and lower values of hyperglycemia were associated with a negative influence, even though the evidence was limited. Zamir et al. [29] reported a significant association with worse neurological outcomes for exposure to a high hyperglycemia threshold—>10 mmol/L—and low hyperglycemia threshold—8 mmol/L—but only if occurring on 3 consecutive days; this shows a possible role for the duration and frequency of exposure, and not only for the absolute value. Similar results were presented by Villamizar at al [28] who found a negative impact on NDI for hyperglycemia >8.2 mmol/L for ≥5 days but not if <5 days. As we previously emphasized, we found a wide heterogeneity of NDI both in terms of definition and how the results of the validated tools were reported. The choice of including mild NDI in the pathological outcome agrees with other studies that investigated neurodevelopment in preterm infants [32]; however, it could reduce specificity and overestimate the impact of hyperglycemia on neurodevelopment.

A second limitation of this review stands in the method of quantifying glycemia, ranging from whole blood [19,27] to plasma [29]: different methods provide different measures [33,34], hence the absence of methodological description may bias the conclusion. 

Our findings agree with other recent reviews on the subject [35,36]: while hyperglycemia negatively impacts short-term clinical outcomes in preterm infants [15], more uncertainties remain about the impact of hyperglycemia on long-term outcomes. The variable definition of a glucose threshold for hyperglycemia in this population contributes to conclusions drawn on this topic. Paulsen et al. [36] underlined the critical need for well-designed studies that could provide stronger evidence on this topic. Unlike Paulsen et al. [36], we focused only on neurological outcome, and we used stricter criteria, excluding studies without a control group. Our review includes two more recent studies with larger sample sizes: 533 patients in Zamir at al. [29] and 280 in Boscarino et al. [30], indicating the growing interest on this topic. Despite this difference, our conclusion did not change: due to the poor quality and heterogeneity of the included studies, we are still unable to provide conclusive recommendations for what is the actual risk of adverse neurological outcomes in preterm infants exposed to hyperglycemia. 

The last significant element to consider is treatment strategy, which was quite heterogeneous among the studies. In clinical practice, the first step is reducing the glucose infusion rate and, secondly, insulin administration [35]. Both these strategies may have a negative influence on neurodevelopment. A decreased glucose intake is often achieved through the reduction of parenteral nutrition, which induces a caloric deficit. The limiting nutrient intake could have unfavorable effects on long-term growth [27] but also on neurodevelopment, as Villamizar et al. [28] found. An adverse effect of insulin therapy was hypoglycemia, which is already in itself a factor of a worse neurological outcome [37]. However, the real risk of hypoglycemia, secondary to insulin, was not well known in preterm infants: Heald et al. [26] did not find a higher risk, while other studies drew the opposite conclusion [38]. Nevertheless, insulin use seemed not to change the survival without NDI in preterm infants, as Tottman et al. [25] and Zamir et al. [29] demonstrated. All the studies included in this review hinted at these controversial issues but each of them in different ways. Therefore, establishing the most effective treatment strategy remains a challenge.

The strength of our review was the rigorous methodology in selecting studies, but there were some limits. Several studies had a limited numerosity with imprecise estimates of effect. Data regarding the neurological assessment were sometimes incomplete and included different assessment tools with a consistent number of dropouts at follow-up assessment. 

Lastly, we cannot exclude other determinants, beside neonatal hyperglycemia, influencing the neurodevelopmental outcome and we are unable to weigh up the individual components in the absence of homogenous and large cohort studies. 

Therefore, future studies should be prospective cohort, randomized trials. Populations should be divided into groups of hyperglycemic and normoglycemic. A shared definition of hyperglycemia should be adopted, with two possible thresholds of 8.2 and 10 mmol/L. The duration of hyperglycemia, nutritional intake and insulin treatment should be considered modifying factors; in addition, several confounding factors should be considered: perinatal characteristics, morbidities of prematurity, episodes of hypoglycemia and socio-demographic characteristics. Neurodevelopment assessment should be performed with validated tools and a shared definition of NDI, including the mild category, should be determined. 

## 5. Conclusions

Best management practices for hyperglycemia in the preterm infant remain unknown despite its common occurrence. Even if this review suggests that neonatal hyperglycemia might negatively influence neurological development, the quality of evidence is poor. The reduction in the heterogenicity of the definition of hyperglycemia and its treatment, and the adoption of standardized neurodevelopment assessment, are necessary to design future studies that could improve the quality of evidence and provide better guidance for clinicians. 

## Figures and Tables

**Figure 1 children-09-01541-f001:**
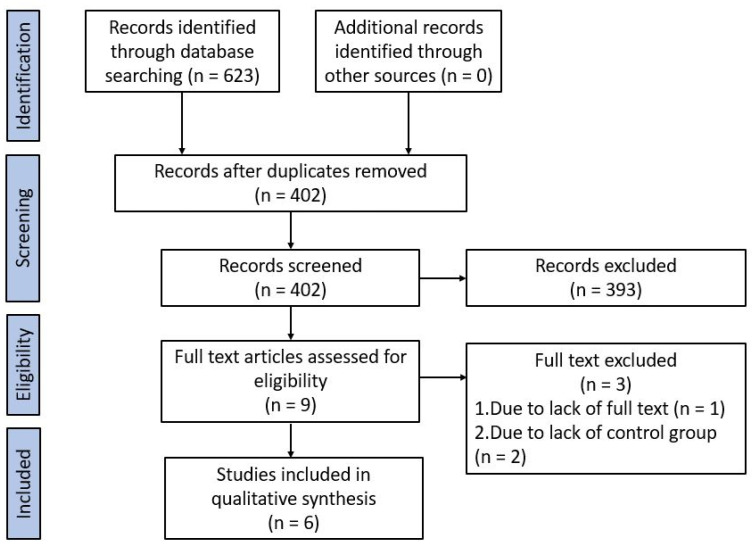
PRISMA flow chart of screening and selection results.

**Table 1 children-09-01541-t001:** Details of included studies.

First Author, Year	Study Type	Place, Time	Population	Sample Size (Incidence of Hyperglycemia)	Definition of Hyperglycemia	Experimental Group/Control Group	Age at Follow-Up	Sample Size at Follow Up (Experimental Group)
Van der Lugt, 2010 [19]	Retrospective	Netherlands, 01.2002–12.2006	≤32 GA	859 (8%)	at least 2 ≥ 10 mmol/L during a 12 h period.	Hyperglycemic/no hyperglycemic	2 y ± 3 m CA	96 (34%)
Heald, 2012 [26]	Retrospective	Australia, 01.2006–12.2008	<29 GA	97 (18%)	≥10 mmol/L and significant glycosuria	insulin/no insulin	1 y CA	73 (12%)
Tottman, 2017 [27]	Retrospective	New Zeland, 07.2005–10.2008	<30 GA/BW <1500 g	360 (20%)	≥8.6 mmol/L on ≥2 measures >1 h apart or any >10 mmol/L during the first 7 DOL	Hyperglycemic/normoglycemic	2 y CA	280 (20%)
Villamizar, 2020 [28]	Retrospective	USA, 02.2012–06.2016	BW <1500 g	97 (48%)	1 or more >8.2 mmol/L	Hyperglycemic/normoglycemic	1 y CA	66 (49%)
Zamir, 2021 [29]	Retrospective	Sweden, 04.2004–03.2007	<27 GA	533 (69%)	Four categories: >8–10–12–14 mmol/L at least 1 or 2–3 consecutive days during the first 28 DOL	Hyperglycemic/no hyperglycemic	6.5 y ± 3 m	436 (87%)
Boscarino, 2021 [30]	Prospective	Italy, 01.2015–12.2019	<32 GA/BW <1500 g	280 (29%)	2 consecutive >10 mmol/L at least 3 h apart	Hyperglicemic/no hyperglicemic	2 y CA	108 (30%)

BW, birth weight; y, year; m, month; CA, corrected age.

**Table 2 children-09-01541-t002:** Neurodevelopment outcome.

Neurodevelopmental Assessment Tool	Risk of Neurological Impairment in the Hyperglycemic Population					
	Van der Lugt [19]	Heald [26]	Tottman [27]	Villamizar [28]	Zamir [29]	Boscarino [30]
BSID		↔	↑ *	↑ **		↑ ***
GMDS		↔				
WISC					↑	
MABC					↑	
CP		↔	↑ *		↔	
B or D		↔	↑ *		↔	
Other	↑					

CP cerebral palsy; B blindness; D deafness; ↑ higher risk of worse outcome in experimental group; ↔ no significant difference in the outcome; not analyzed; * not confirmed after adjustment for GA, BW z-score and socioeconomical level; ** same results after adjustment for Score for Neonatal Acute Physiology–II but not for first-week nutrition intake; ***confirmed after adjustment for confounders.

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
