# Peer review of "Neonatal Hyperglycemia and Neurodevelopmental Outcomes in Preterm Infants: A Review"

_children, 2022, doi:10.3390/children9101541_

Round 1

Reviewer 1 Report

I appreciate the efforts of authors in systematically reviewing the impact of hyperglycemia in preterm infants on their neurodevelopment. This is a well written paper in general. I have following comments

1   In the abstract the authors say they used BSID and GMDS for the first 5 years but in the inclusion criteria subsection of methods they say they used BSID and GMDS for 2 years.

    Authors use BSID and BSDI for Bayley Scale of Infant Development. A consistency should be maintained in the abbreviations.

          Authors defined NDI as scores < -1 SD for BSID, GMDS and WISC. Was this based on how the six studies included in this review defined NDI? Many investigators use > 2SD below the normal for defining abnormality. A clarification will be helpful.

There are a couple of grammatical errors in the manuscript. E.g. line 45 should say cohort ‘studies and case control studies’. Line 193 should say ‘all the studies’. A few sentences across the manuscript need better construction.

Author Response

Dear Reviewer, thank you for your careful revision. Following your suggestions we made these changes:

  • There was a mistake in the methods: as correctly written in the abstract, we considered BSID and GMDS for the first 5 years of life, knowing that BSID investigate up to 3.5 years. For the following age we considered the other tools mentioned.
  • We apologized for the wrong use of the abbreviation; we revised the manuscript adopting the BSID for “Bayley Scale of Infant Development”
  • As regard NDI, during the step of identifying method of our study, we decided to include research that used validated tools to have a standard method simpler to match. However, each included study used different definition of NDI derived from integration of several clinical aspects among which validated tools as BSID, GMDS, WISC and MABC, were considered but not the only one. Furthermore, each studies reported the results of the tools in different way: as z-score or as median of the whole population and some studies (Zamir et al and Tottman et al) categorized NDI into different level of seriousness based on results of these validated tools: the mild category coincided to score < -1SD and moderate/severe category to score < -2SD. Therefore, we had to find a definition of NDI that could include all possible level of NDI. All the validated tools considered moderate/severe NDI a score <-2 SD but more uncertainty existed on the interpretation of score between -2 SD and -1 SD: this range defined the mild category but several studies showed a tendency to underestimate NDI, especially for the BSID. Therefore we decided to include the mild category as a pathological outcome, knowing that this decision could overestimate the impact of hyperglycemia on neurodevelopment. We also underlined how to report the results of tools was an important point: the heterogeneity of the included studies prevented from drawing strong conclusions, therefore future studies should use a shared mode of report results.
  • We corrected the grammatical errors and tried to better construct other sentences to improve the reading.

Reviewer 2 Report

Guidici et al desribe the results of 6 seleted studies evaluating the neurodevelopmental outcome of preterm infants with hyperglycemia. With the exception of one study investigation was done retrospectively. The overall follow up rate ist low (48%). It is highlighted that defintion of hyperglycemia is not standardized likewise the follow up examinations. They found an assoziation of hyperglycemia with neurological delay but the evidence is low and far away from proof of causality. Nevertheless, the results may help to bear in mind that hypergylcemia may worsen outcome.

Given the low quality of evidence the authors are concise an careful in drawing conclusions which is appropriate. However, the last sentence in the abstract should also be more reserved regarding consequences of hyperglycemia. The suggestion for better designed future studies is welcome.

Author Response

Dear Reviewer, thanks for your accurate revision and for your kind suggestions. As you underlined our review has several limits due to the extreme heterogeneity of the studies currently available.

As you highlight, the last sentence in the abstract could be too imprecise because we didn’t clearly talk about the quality of evidence. Therefore, we changed the sentence specifying the low quality of evidence and underlining that further studies are necessary to strengthen the suggestion.

Following your advice, we added a paragraph at the end of the discussion where we tried to give a suggestion on how future studies should be designed. Actually in our review we often underlined that the studies included had different criteria, adopted different definition of hyperglycemia and treatment strategy and then also the neurodevelopment assessment was quite disomogeneous. Therefore it was important to give a common way for future study so that we will have better quality of evidence.

Reviewer 3 Report

In the manuscript „Neonatal Hyperglycemia and Neurodevelopmental Outcomes in Preterm: a Review“ Guiducci and colleagues investigated the impact of hyperglycemia on neonatal neurodevelopment by intensive literature research. In total, Guiducci et al. identified 623 studies by screening scientific databases. After narrowing down the selection, Guidicce and colleagues included six studies in their analyses.

Due to parenteral feeding, hyperglycemia is a common problem in neonatal intensive care and can be related to gestational age. Frequently, increased glucose concentrations have to be treated by insulin administration.

Overall, this review discusses an important clinical question, what are the long-term effects on the brain caused by neonatal hypercalcemia in preterm infants.

The authors have already discussed limitations. It is particularly important to consider the heterogeneity of the studies as well as the small number of participants in some studies.

Here are my comments to the authors:

The introduction is very brief. Please include reasons for neonatal hypercalcemia in the introduction part.

The description of the analyzed studies is to short. Give detailed information about analyses, groups and possible treatments.

Was there an insulin treatment?

Authors should discuss a possible second hit mechanism by insulin-induced hypoglycemia on neurodevelopment after hyperglycemia treatment.

Author Response

Dear Reviewer, thanks for your accurate revision and for your kind suggestions. We answer as follows:

  • As you suggest for the introduction, we added some details about the pathogenesis of hyperglycemia focusing on preterm infants, we analysed the two main causes: reduced insulin secretion and relative insulin resistance.
  • In the first part of the result, we added more details for each study, specifying the study design with focus on the definition of hyperglycemia, the distinction in different group base on glycemia, the characteristics of follow up assessment
  • Treatment strategy of hyperglycemia is a critical point as hyperglycemia itself, and we thank you for the focus. We added details about the different treatment strategy adoped, underlining where insulin was used and where not. The first step of reducing glucose infusion rate and secondarily starting insulin administration were the common strategy adopted.
  • As regards insulin-induced hypoglycemia, this is another crucial point. More evidence exist about the negative role of hypoglycemia on long term outcome, but there are no strong evidence from the literature about the real incidence of hypoglycemia secondary to insulin in preterm infants, with conflicting opinions. Consequently, it is not clear if this added episodes of hypoglycemia could be a potential negative factor of worsen neurodevelopment or if hyperglycemia by itself is responsible of neurodevelopment delay.

Round 2

Reviewer 3 Report

The authors addressed my concerns